# Demonstration of Co-Infection and Trans-Encapsidation of Viral RNA In Vitro Using Epitope-Tagged Foot-and-Mouth Disease Viruses

**DOI:** 10.3390/v13122433

**Published:** 2021-12-03

**Authors:** Kay Childs, Nicholas Juleff, Katy Moffat, Julian Seago

**Affiliations:** The Pirbright Institute, Ash Road, Woking GU24 0NF, Surrey, UK; kay.childs@pirbright.ac.uk (K.C.); Nick.Juleff@gatesfoundation.org (N.J.); katy.moffat@pirbright.ac.uk (K.M.)

**Keywords:** foot-and-mouth disease, FMDV, co-infection, trans-encapsidation

## Abstract

Foot-and-mouth disease, caused by foot-and-mouth disease virus (FMDV), is an economically devastating disease affecting several important livestock species. FMDV is antigenically diverse and exists as seven serotypes comprised of many strains which are poorly cross-neutralised by antibodies induced by infection or vaccination. Co-infection and recombination are important drivers of antigenic diversity, especially in regions where several serotypes co-circulate at high prevalence, and therefore experimental systems to study these events in vitro would be beneficial. Here we have utilised recombinant FMDVs containing an HA or a FLAG epitope tag within the VP1 capsid protein to investigate the products of co-infection in vitro. Co-infection with viruses from the same and from different serotypes was demonstrated by immunofluorescence microscopy and flow cytometry using anti-tag antibodies. FLAG-tagged VP1 and HA-tagged VP1 could be co-immunoprecipitated from co-infected cells, suggesting that newly synthesised capsids may contain VP1 proteins from both co-infecting viruses. Furthermore, we provide the first demonstration of trans-encapsidation of an FMDV genome into capsids comprised of proteins encoded by a co-infecting heterologous virus. This system provides a useful tool for investigating co-infection dynamics in vitro, particularly between closely related strains, and has the advantage that it does not depend upon the availability of strain-specific FMDV antibodies.

## 1. Introduction

Foot-and-mouth disease virus (FMDV) is the etiologic agent of foot-and-mouth dis-ease (FMD), an economically important vesicular disease affecting even-toed ungulates including critical livestock species such as cattle, sheep and pigs. FMDV is a member of the Aphthovirus genus within the Picornaviridae family of viruses and is comprised of a positive-sense, single-stranded RNA genome of around 8.5 kb inside a small, non-enveloped icosahedral capsid. The genome contains a single open reading frame encoding a polyprotein that is post-translationally processed by viral proteases into the mature viral structural and non-structural proteins [1].

The FMDV capsid contains 60 copies of each of the structural proteins VP1 (1D), VP2 (1B), VP3 (1C) and VP4 (1A), encoded by the N-terminal P1 region of the polyprotein. VP2 and VP4 are initially synthesised as a precursor (VP0) which combines with a single copy each of VP1 and VP3 to form a protomer. In a process involving sequential self-assembly of subunits, 5 protomers form a pentamer and 12 pentamers form the capsid encasing a single copy of the RNA genome [2]. Finally, VP0 is cleaved into VP2 and VP4, and while VP1, VP2 and VP3 are exposed on the surface of the capsid, VP4 is entirely internal. Virus entry into cells is mediated by viral attachment to cell surface integrin receptors, that include αvβ1/β3/β6/β8, through an RGD motif located within the GH loop of VP1 [3,4,5,6]. The GH loop is a highly flexible loop formed by amino acids 140–160 of VP1 which protrudes from the surface of the capsid to facilitate interaction with receptors.

FMDV exists as seven different serotypes, A, O, C, Asia 1, Southern African Territories (SAT)1, SAT2 and SAT3 which exhibit distinct, but partially overlapping, geographical distributions. Considerable genetic diversity also exists within each serotype, and antibodies produced by infection or vaccination with one strain do not necessarily provide effective protection against another strain even within the same serotype. This significantly impacts upon the success of vaccination campaigns in the field, which is the main control measure in countries where FMD is endemic. Furthermore, the emergence of new strains that escape existing immunity is a persistent threat to global efforts to suppress this disease.

In areas where FMDV is highly prevalent and several serotypes co-circulate, co-infection with two (or more) genetically distinct viruses is not uncommon. The three SAT serotypes are endemic in Southern Africa where they co-circulate amongst wild African buffaloes (*Syncerus caffer*) which act as the primary natural reservoir of FMDV and are a source of outbreaks in cattle. It has long been appreciated that individual buffalo may harbour two, or even three different SAT serotypes at one time, and may remain persistently infected for extended periods of up to several years [7,8]. Many examples of mixed serotype infections in cattle have also been documented [9,10,11,12], and more than one strain of the same serotype has been identified within a single sample [12]. Co-infection with antigenically diverse strains is of particular concern due to the increased risk associated with the generation of novel recombinant viruses with the potential to evade immune responses raised against previously circulating strains. Indeed, a recent study has demonstrated that cattle can be experimentally co-infected with serotype A and serotype O FMDVs, and that this can give rise to recombinant viruses [13] even when infection with the two viruses is staggered by several weeks. This highlights the importance of persistently infected carrier animals as important potential sources of recombinants in the field.

Recombination occurs frequently between closely related genotypes [14], and although less common, inter-serotype recombinants have also been isolated [15,16,17,18,19]. Analysis of recombination breakpoints has revealed “hotspots” of recombination at the boundaries of the P1 region, with breakpoints within the VP1–3 region that encodes the capsid proteins being rare [20]. This is a consequence of the extensive interactions between proteins within the capsid placing significant constraints on the compatibility of proteins de-rived from different virus strains to form a functional capsid. At the genetic level, recombination events occur at a similar rate across the genome, however, recombinants that contain mixed capsid proteins are likely to have impaired fitness relative to the parental genomes and fail to propagate as a result. This has led to the capsid proteins evolving as a separate unit, with recombinant viruses inheriting the capsid protein genes as a single block. Nevertheless, recombination events within the capsid region have been described [21] and have the potential to give rise to novel viruses that escape existing immunity.

Co-infection of cells with wild-type FMDV strains, particularly strains that are closely related, is difficult to study due to the lack of strain-specific or even serotype-specific anti-bodies. To overcome this limitation, we have utilised recombinant FMDVs with HA and FLAG epitope tags inserted into VP1 to study co-infection in vitro. We hypothesised that co-infection may lead to the production of chimeric capsids containing subunits encoded by both co-infecting viruses, and that trans-encapsidation of one genome by capsids encoded by the other genome may be observed. Both phenomena have implications for immune evasion due to incomplete antibody-mediated neutralisation and “shielding” of a viral genome in a heterologous capsid. This in vitro system has the potential to facilitate the understanding of co-infection dynamics, capsid compatibility and the factors governing trans-encapsidation, which all have implications for the emergence of novel viral strains in the field, and importantly does not rely on the availability of strain-specific FMDV antibodies.

## 2. Materials and Methods

### 2.1. Cells and Viruses

ZZ-R 127 goat epithelium cells were maintained in Dulbecco’s modified Eagle’s medium/Ham’s F12 medium (Merck Life Science, Dorset, UK) with 10% foetal bovine serum (FBS) and penicillin/streptomycin (Sigma) at 37 °C in 5% CO_2_. The generation of HA- and FLAG-tagged FMDV O1K/O UKG35 has been previously described [22]. An infectious copy plasmid encoding an FMDV Asia 1 Bahrain/O1Kaufbeuren (O1K) chimera was constructed using reverse genetics as described previously [23]. Briefly, cDNA encoding the VP2, VP3, VP1 and 2A proteins was removed from an existing O1K infectious copy plasmid leaving cDNA encoding the Lpro, VP4, 2B, 2C, 3A, 3B, 3C, 3D proteins. The removed cDNA was replaced with the corresponding Asia 1 Bahrain cDNA from a pGEM9zf subclone encoding the HA (YPYDVPDYA) epitope tag in the GH loop of VP1 between valine 153 and serine 154. DNA encoding the peptide tag was inserted into the subclone by performing two consecutive rounds of PCR amplification using a QuikChange Lightning Mutagenesis kit (Agilent Technologies, Stockport, UK) with the following primers according to the manufacturer’s instructions: Asia1 HA-F1: 5′-CATACGACGTACCAGATTACGCTAGCAGGCAACTGC-3′; Asia1 HA-R1: 5′-AGCGTAATCTGGTACGTCGTATGCACCCTTTGTGCA-3′; Asia1 HA-F2: 5′-TGCACAAAGGGTGTACCCATACGACGTACC-3′; Asia1 HA-R2: 5′-GGTACGTCGTATGGGTACACCCTTTGTGCA-3′. RNA was transcribed from the full-length ‘tagged’ infectious copy plasmid using a MEGAscript T7 kit (Invitrogen, ThermoFisher Scientific, Loughborough, UK) and electroporated into BHK-21 cells as previously detailed [22].

### 2.2. Western Blotting

Proteins were separated by SDS-PAGE on NovexTM WedgeWellTM 4–20% Tris-glycine gels (Invitrogen) and transferred to nitrocellulose membranes (LI-COR, Cambridge, UK). Immunoblotting was carried out using mouse anti-FLAG M2 (Agilent Technologies), rabbit anti-FLAG (OriGene technologies R1180), rabbit anti-HA (Sigma H6908) and mouse anti-HA (Sigma H3663) antibodies to detect the epitope tags, the mouse mAb 2C2 to detect the FMDV 3A protein and 3A-containing precursors [24] and mouse anti-γ-tubulin (Sigma T6557) as a loading control. Bound primary antibodies were detected using IRDye^®^ 680RD goat anti-rabbit IgG and IRDye^®^ 800CW goat anti-mouse IgG secondary antibodies (LI-COR) and visualised on an Odyssey CLx imaging system (LI-COR).

### 2.3. Immunofluorescence

Cells cultured on glass coverslips were infected with HA- or FLAG-tagged FMDV O1K/O UKG35 or co-infected with both. After 4 h, cells were washed in PBS, fixed with 4% paraformaldehyde, permeabilised with 0.2% *v*/*v* Triton-X100, washed and blocked with 0.5% BSA in PBS. Cells were then incubated with mouse anti-HA (Sigma H3663, IgG1), rabbit anti-FLAG (OriGene technologies R1180) and mouse mAb 2C2 (IgG2a) [24] antibodies. Labelled cells were washed and then incubated with anti-mouse IgG1-Alexa Fluor (AF) 488, anti-rabbit IgG-AF 568 and anti-mouse IgG2a-AF 633 secondary antibodies (Molecular Probes, Fisher scientific UK Ltd.). Nuclei were stained with DAPI (Merck Life Science) and data were collected using a Leica SP2 laser scanning confocal microscope.

### 2.4. Flow Cytometry

Infected cells were washed in PBS, fixed with 4% paraformaldehyde for 30 min, washed again and permeabilised in FACS diluent (0.2% saponin, 1% BSA in PBS) for 5 min. Cells were then incubated with mouse anti-HA (Sigma H3663) and rabbit anti-FLAG (OriGene technologies R1180) antibodies in FACS diluent for 30 min, then washed three times in FACS diluent before labelling with goat anti-mouse IgG1-AF 488 and goat anti-rabbit IgG-AF 405 (Molecular probes, Fisher scientific UK Ltd., Loughborough, UK) for 30 min in the dark. Labelled cells were washed three times in FACS diluent and resuspended in PBS. Data were collected using DIVA 8 acquisition software and an LSR Fortessa (BD Biosciences, Wokingham, UK) and analysed in FCS express. A minimum of 10,000 events were collected for each sample. Samples were gated on cells (SSC-A vs. FSC-A) and singlets (SSC-A vs. SSC-H), and virus infection was identified as cells positive for HA-AF 488 and/or FLAG-AF 405. Single colour controls were used for compensation, untagged virus was used to set thresholds and single tagged virus infected cells were labelled with both antibodies to determine non-specific tag antibody interactions.

### 2.5. Immunoprecipitation

Confluent monolayers of ZZ-R 127 cells in T175 flasks were either co-infected with HA- and FLAG-tagged FMDV O1K/O UKG35 (MOI = 1/virus) or infected with each virus separately (MOI = 2). About 7.5 h after infection, cells were subjected to a single freeze/thaw cycle and the lysates were clarified by centrifugation. Supernatants were incubated with anti-FLAG^®^ M2 affinity gel (A2220 Merck Life Science UK Ltd., Dorset, UK) or EZviewTM Red anti-HA affinity gel (E6779 Merck Life Science UK Ltd.) for 1 h at room temperature with continual rotation. Beads were washed five times with PBS containing 0.05% Tween-20, and then divided into two equal aliquots for subsequent Western blotting and RT-PCR analysis.

### 2.6. RT-PCR

RNA was purified from immunoprecipitates using RNAzol RT (Merck Life Science UK Ltd.). First strand cDNA was synthesised from RNA templates using RevertAid Re-verse Transcriptase (Fisher Scientific, Loughborough, UK) and an FMDV-specific primer (5′-CCCTCTTCATGCGGTAAAGC-3′). Amplification of HA- and FLAG-containing FMDV sequences was achieved by performing PCR with reverse primers specific for the HA (5′-GCGTAATCTGGTACGTCGTATGGGTACG-3′) or FLAG (5′-GCCTTATCGTCATCGTCTTTGTAGTCC-3′) tags and a common forward primer (5′-CTTGCACTGCCTTACACGGC-3′). Products were analysed by agarose gel electropho-resis.

## 3. Results

### 3.1. Co-Infection of Cells with Epitope-Tagged Viruses

We have previously described the construction of epitope-tagged FMDVs containing either an HA or a FLAG tag located within the GH loop of VP1 (shown schematically in Figure 1a) [22]. The GH loop is a highly flexible, surface-exposed loop which contains the RGD motif responsible for binding to integrin receptors on the cell surface. It also shows considerable natural variation among isolates and can tolerate the insertion of exogenous sequences. Our data showed that epitope-tagged viruses retained the ability to bind αvβ6 integrin receptors, replicated with similar kinetics to field strains and the parental untagged virus, and produced similar plaque morphologies.

To investigate the feasibility of using these viruses to study co-infection dynamics and trans-encapsidation events in FMDV, we first ascertained whether cells could be simultaneously co-infected with both FLAG- and HA-tagged viruses. Cultures of goat epithelium cells, which express the FMDV integrin αvβ6, were infected with tagged viruses individually or in combination, and tagged VP1 proteins were detected by Western blotting of whole cell lysates using anti-HA and anti-FLAG antibodies (Figure 1b). FLAG-VP1 and HA-VP1 were present in lysates from cells infected with FLAG-FMDV and HA-FMDV respectively, and both were present in lysates from co-infected cells. To demonstrate that co-infection occurs at the single cell level, infected cells were also analysed by immunofluorescence microscopy (Figure 1c). In cell cultures that were co-infected with HA-FMDV and FLAG-FMDV, cells that were positive for both tags were identified, confirming that individual cells can be simultaneously co-infected with FLAG-FMDV and HA-FMDV. Staining with an antibody directed against the non-structural protein 3A and the uncleaved 3A/3B precursors confirmed that virus replication was also taking place in infected cells.

To confirm and quantify the level of virus co-infection, cells infected with either one or both epitope-tagged viruses were analysed by flow cytometry. Analysis of cell populations infected with a single epitope-tagged virus at an MOI of 1 showed that 36.2% of the population was infected with FLAG-FMDV, and 18.19% of the population was infected with HA-FMDV (Figure 2). In populations of cells co-infected with both FLAG-FMDV and HA-FMDV at an MOI of 0.5 per virus, 10.2% of the population was infected only with FLAG-FMDV, 17.82% was infected only with HA-FMDV and 2.37% was co-infected FLAG-FMDV and HA-FMDV (Figure 2). Co-infected cells were detected in two further separate experiments, although variation in the percentage of cells infected with each virus was observed (Appendix A). 

### 3.2. Co-Infection with Epitope-Tagged FMDVs of Different Serotypes 

Having demonstrated that epitope-tagged viruses of the same serotype can co-infect cells, we next wanted to examine co-infection with FMDVs of different serotypes. To facilitate these experiments, an HA-tagged Asia 1/O1K chimeric virus was generated by removing the DNA encoding the external capsid proteins from an FMDV O1K infectious copy plasmid and replacing it with DNA encoding the external capsid proteins from an Asia 1 (Bahrain) strain. The position of the tag is shown schematically in Figure 3a. The virus produced from this infectious copy plasmid has a capsid comprised of the Asia 1 serotype VP1, VP2 and VP3 proteins, and the remaining viral structural and non-structural proteins from O1K. Expression of the HA-tagged Asia 1 VP1 was confirmed by immunoblotting of whole cell lysates prepared from goat epithelium cells infected with HA-FMDV Asia 1/O1K (Figure 3b), and virus replication was demonstrated by blotting with an antibody to 3A and 3A/3B precursors. HA-tagged Asia1/O1K displayed comparable growth kinetics and plaque morphology to untagged Asia1/O1K (Appendix A). 

To study co-infection with Asia 1 and O serotype viruses, goat epithelium cells were infected with HA-FMDV Asia 1/O1K and FLAG-FMDV O1K/O UKG35 either individually or together. Flow cytometry analysis of single infections (MOI = 1) showed that 32.81% of cells in the population were infected with FLAG-FMDV O1K/O UKG35, and 66.44% of cells were infected with HA-FMDV Asia 1/O1K (Figure 3c). In the cultures of cells infected with both viruses (MOI = 0.5 per virus), 48.07% of cells were infected with HA-FMDV Asia 1/O1K alone, 5.32% of cells were infected with FLAG-FMDV O1K/O UKG35 alone and 4.99% of cells were co-infected with both viruses (Figure 3c).

### 3.3. Generation of Chimeric Capsids

Co-infection of a single cell with two different viruses may result in the generation of chimeric capsids containing mixtures of protein subunits encoded by both genomes. To examine whether this occurs during co-infection with HA-FMDV and FLAG-FMDV, viruses produced from co-infected cells were immunoprecipitated using agarose beads coated with either HA- or FLAG-specific antibodies, and then analysed by Western blotting to detect HA-VP1 in the FLAG immunoprecipitates and FLAG-VP1 in the HA immunoprecipitates (shown schematically in Figure 4a). As a control, HA- and FLAG-tagged viruses that had been grown in separate cell cultures and subsequently mixed were subjected to identical immunoprecipitation procedures to eliminate the possibility that a positive signal could arise from HA- and FLAG-tagged viruses adhering to each other during purification.

These experiments showed that HA-tagged VP1 was successfully immunoprecipitated from both co-infections and from combined lysates from separate infections using anti-HA-agarose beads (Figure 4b, left hand side). Interestingly, FLAG-VP1 was present in HA-immunoprecipitates from co-infected cells, but not in HA-immunoprecipitates from separate, combined lysates or when no HA-tagged virus was present. In the reciprocal experiment, FLAG-tagged VP1 was immunoprecipitated from both co-infections and separate, combined infections using anti-FLAG-agarose beads (Figure 4b, right hand side), and this was accompanied by HA-tagged VP1 in the FLAG-immunoprecipitates from co-infected cells, but not from combined lysates from separate infections or when no FLAG-tagged virus was present (note that the HA-tagged VP1 can be observed below the antibody light chain labelled * in the right hand panel of Figure 4b). These experiments provide evidence that supports the hypothesis that capsid proteins encoded by different incoming viral genomes may be incorporated into a single chimeric progeny virus particle during co-infection.

### 3.4. Trans-Encapsidation of FMDV Genomes

Trans-encapsidation refers to the packaging of a viral genome into a capsid that has been produced from a heterologous, co-infecting virus. Although this process has been studied extensively in poliovirus (PV), there are no reports of trans-encapsidation during co-infection of cells with two strains of FMDV. To investigate whether trans-encapsidation occurs in cells co-infected with HA-FMDV and FLAG-FMDV, viral RNA was isolated from separate aliquots of the immunoprecipitated material described above and subjected to RT-PCR analysis using primers specific for either the HA- or FLAG-tagged VP1 sequences. As expected, in HA immunoprecipitates from both co-infected cells and from the combined lysates from separate infections, a 149 bp PCR product was detected using the HA-VP1-specific primers (Figure 4c, left hand side). Crucially, use of the FLAG-specific primers revealed the presence of FLAG-VP1-encoding genomes in the HA immunoprecipitate from co-infected cells but not from the combined lysates from separate infections or in the absence of the HA-tagged virus. Similarly, FLAG immunoprecipitates from co-infected cells and from combined lysates from separate infections both yielded a PCR product with the FLAG-specific PCR primers, but only co-infected cells contained HA-VP1-encoding genomes (Figure 4c, right hand side). These data confirm that trans-encapsidation of a viral genome into a capsid derived from a co-infecting FMDV can occur and can be detected using this method.

## 4. Discussion

Picornavirus replication takes place on membranous structures formed during ex-tensive remodelling of the cytoplasmic contents following infection. The origins of the membranes that form these replication centres are not entirely clear, but they may, at least in PV-infected cells, derive from vesicles trafficking between the ER and the Golgi [25]. Electron microscopy studies of FMDV-infected cells revealed a cytoplasmic reorganisation distinct from cells infected with other picornaviruses that showed a complete redistribution of organelles to one side of the nucleus where the replication centre was located [26]. Within these replication centres, VP0, VP1 and VP3 produced from the same polyprotein remain associated after cleavage and form a protomer coincident with their synthesis. High local concentrations of protomers produced from the same genome favours the formation of pentamers and capsids containing homologous capsid proteins. For the same reason, newly synthesised genomes are most likely to be packaged by capsid proteins produced from the same parental genome. PV studies have shown that, in addition to physical compartmentalisation, there is a strong coupling between translation, RNA replication and packaging which ensures that only RNAs that have been successfully translated can be templates for RNA replication to produce genomes for incorporation into new virus particles [27,28]. This acts as a quality control mechanism to prevent encapsidation of defective RNA genomes which fail to produce functional proteins. 

During co-infection, the generation of chimeric capsids, trans-encapsidation of heterologous genomes and recombination require some degree of mixing between replication complexes. Although this is clearly not an infrequent event, the efficiency of homologous encapsidation was found to be 2 orders of magnitude higher than heterologous encapsidation in cells co-infected with two different types of PV [29]. The dynamics of mixing between components derived from different incoming genomes is difficult to study, but it may be envisaged that at later times post-infection with FMDV, when the replication centre takes up most of the cytosol [26], the concentrations of viral components increases the chances of mixing, thus raising the likelihood of the generation of chimeric particles.

Trans-encapsidation has been studied extensively in PV using replicon systems in which the sequences encoding the PV capsid proteins have been replaced with a reporter gene. In these systems, the in vitro transcribed replicon RNA is transfected into cells previously infected with a helper virus encoding capsid proteins that can package the replicons into infectious particles. These experiments demonstrated that PV replicons can be efficiently packaged by capsid proteins from other types of PV, but not by the capsid proteins from other species of picornavirus [29,30,31]. This, along with the exclusion of negative sense viral RNAs and cellular RNAs from incorporation into virus particles, is indicative of some degree of specificity during the selection of genomes for encapsidation. However, it has proven difficult to identify bona fide RNA packaging signals in picornavirus genomes, and to date the only member of this family known to contain such an element is Aichi virus in which a stem-loop in the 5′-UTR is necessary for successful packaging [32]. One reason that it may have been difficult to identify packaging signals is that they may be dispersed throughout the genome and be comprised of structural elements rather than specific RNA sequences that make numerous contacts with the inside of the capsid [33,34].

Trans-encapsidation can have important consequences for viral pathogenesis since it may allow viral genomes that have been packaged into heterologous capsids to escape immunity in the next host. An early example of this was reported by Trautman and Sut-moller who described the “genomic masking” of an FMDV RNA inside a bovine entero-virus (BEV) capsid [35]. Importantly, they showed that the chimeric particle could be neutralised by BEV-antiserum, but not by FMDV-antiserum. Similarly, capsids containing mixtures of capsid proteins from different strains of FMDV could result in only partial neutralisation by antisera. In our experiments, we cannot exclude the possibility that viral genomes were packaged in mixed capsids containing both HA and FLAG-tagged units, and therefore not solely in capsids derived from the heterologous co-infecting virus. However, there is no clear mechanistic basis for the selection of mixed capsids over those derived from the heterologous, co-infecting virus.

The emergence of viruses with chimeric capsids, masked genomes and novel recombinant capsids poses a threat to the livestock industry due to the risk of immune escape from vaccine control and subsequent outbreak potential. There is consequently a pressing need to understand the processes driving the generation of viral diversity and the contribution of co-infection to bringing new combinations of genes together. Here we have shown that using viruses with epitope-tagged VP1 proteins in their capsids is a viable method to study co-infection of cells with different strains of FMDV, from the same or from different serotypes. This technique allows the use of widely available antibody-conjugated agarose beads to isolate and characterise viruses produced from such co-infection experiments. We show that FLAG-VP1 and HA-VP1 can be co-precipitated from cells co-infected with FLAG-FMDV O1K/O UKG35 and HA-FMDV O1K/O UKG35, indicating that VP1 proteins from both viruses can be incorporated into the same pentamers or capsids. This system could be used to study compatibility between capsid proteins from different strains without the requirement for serotype-specific antibodies, and to investigate the factors influencing trans-encapsidation. Other applications could include the study of competition between co-infecting viruses, with emphasis on the effects of MOI and different cell types. Serial passage of virus from mixed infections of SAT1, SAT2 and SAT3 in cell culture showed that SAT1 outcompeted the other two serotypes [36]. This analysis was carried out by RT-PCR analysis of samples taken from each passage, but with the epitope-tagged viruses, flow cytometry could be used to monitor the percentage of cells infected by each virus. An important aspect of this approach is that by introducing distinct epitope tags into the capsids of two different viruses it is possible to investigate co-infection dynamics in cells infected with two closely related viruses which are otherwise immunologically indistinguishable. Since most successful recombination events occur between closely related strains this will provide a useful tool for such analyses. 

## Figures and Tables

**Figure 1 viruses-13-02433-f001:**
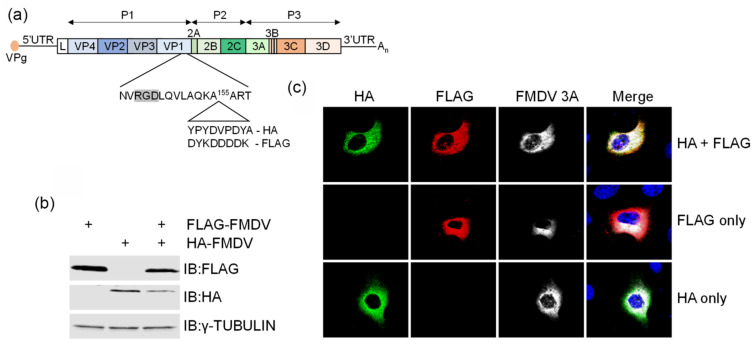
Co-infection of goat epithelium cells with HA- and FLAG-tagged FMDV O1K/O UKG35. (**a**) Schematic diagram showing the position of the tag insertion in the FMDV O1K/O UKG35 infectious clone. (**b**) Confluent monolayers of ZZ-R 127 cells were infected with FLAG-FMDV O1K/O UKG35 (MOI = 1), HA-FMDV O1K/O UKG35 (MOI = 1) or both (MOI = 0.5/virus) for 6 h. Epitope-tagged VP1 proteins in lysates from infected cells were detected by immunoblotting (IB) with antibodies to the FLAG and HA tags, and an anti-γ-tubulin antibody was used to demonstrate equal loading. (**c**) Immunofluorescence microscopy of cells infected with FLAG-FMDV O1K/O UKG35 (MOI = 1), HA-FMDV O1K/O UKG35 (MOI = 1) or both (MOI = 0.5/virus). Cells were stained with antibodies to the HA tag (AF 488-green), the FLAG tag (AF 568-red), FMDV 3A and 3A/3B precursors (AF 633-white) and nuclei were stained with DAPI (blue).

**Figure 2 viruses-13-02433-f002:**
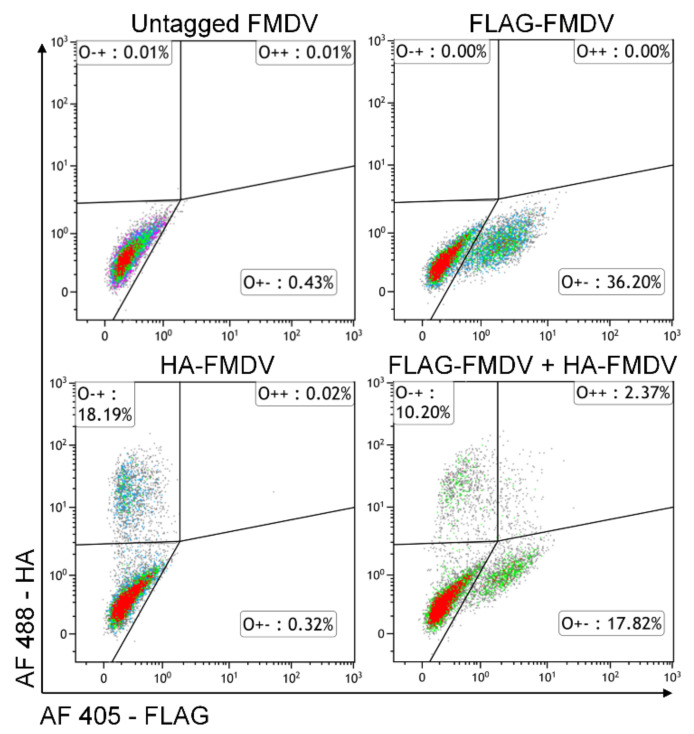
Flow cytometry analysis of goat epithelium cells infected with HA- and FLAG-tagged FMDV O1K/O UKG35. Cells were infected with untagged FMDV O1K/O UKG35 (MOI = 1), FLAG-FMDV O1K/O UKG35 (MOI = 1), HA-FMDV O1K/O UKG35 (MOI = 1), or co-infected with FLAG-FMDV O1K/O UKG35 and HA-FMDV O1K/O UKG35 (MOI = 0.5/virus) for 5 h. Cells were fixed, permeabilised and labelled with anti-HA and anti-FLAG antibodies and analysed by flow cytometry. The percentage of cells positive for HA only (O-+), FLAG only (O+-) and HA plus FLAG (O++) are shown.

**Figure 3 viruses-13-02433-f003:**
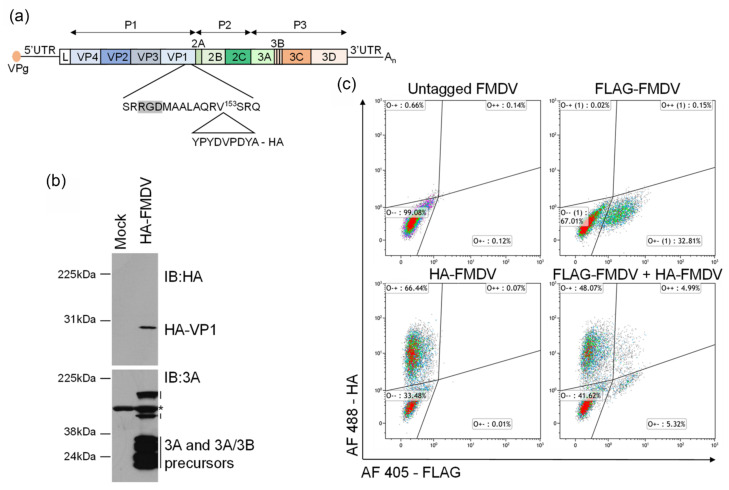
Co-infection of goat epithelium cells with HA-FMDV Asia 1/O1K and FLAG-FMDV O1K/O UKG35. (**a**) Schematic diagram showing the position of the tag insertion in the FMDV Asia 1/O1K infectious clone. (**b**) Confluent monolayers of cells were mock infected or infected with HA-FMDV Asia 1/O1K (MOI = 1). Whole cell lysates were analysed for the presence of HA-tagged VP1 by immunoblotting (IB) with an antibody to the HA tag, and non-structural protein 3A (and 3A/3B precursors) with the 2C2 mAb. * indicates a non-specific cross-reacting cellular protein which demonstrates equal loading. (**c**) Cells were infected with untagged FMDV O1K/O UKG35 (MOI = 1), FLAG-FMDV O1K/O UKG35 (MOI = 1), HA-FMDV Asia 1/O1K (MOI = 1), or co-infected with FLAG-FMDV O1K/O UKG35 and HA-FMDV Asia 1/O1K (MOI = 0.5/virus) for 5 h. Cells were fixed, permeabilised and labelled with anti-HA and anti-FLAG antibodies and analysed by flow cytometry. The percentage of cells positive for HA only (O-+), FLAG only (O+-) and HA plus FLAG (O++) are shown.

**Figure 4 viruses-13-02433-f004:**
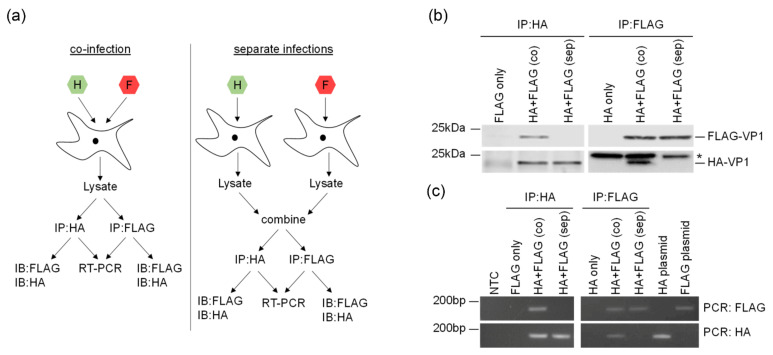
Co-infection results in chimeric capsids and trans-encapsidation of viral genomes. (**a**) Schematic diagram outlining the workflow for the immunoprecipitation experiments. Cells were either co-infected with HA-FMDV O1K/O UKG35 (H) and FLAG-FMDV O1K/O UKG35 (F) or infected with each virus separately. Lysates from co-infected cells were immunoprecipitated using anti-HA affinity gel (IP:HA) or anti-FLAG affinity gel (IP:FLAG), and each immunoprecipitate was split into two aliquots for immunoblot analysis (IB) and RT-PCR. Lysates from cells separately infected with either HA- or FLAG-tagged FMDV were mixed 1:1 and subjected to the same procedure. Lysates from single infections were also used as controls. (**b**) HA and FLAG immunoprecipitates from single infections, co-infections (co) or combined lysates from separate infections (sep) were analysed by immunoblotting for the presence of FLAG- and HA-tagged VP1. * = IgG. (**c**) HA and FLAG immunoprecipitates from single infections, co-infections (co) or combined lysates from separate infections (sep) were subjected to RT-PCR using FLAG- or HA-specific primer sets. Infectious copy plasmids containing HA- or FLAG-tagged viral sequences served as controls. NTC = no template control.

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
