# Peer review of "Demonstration of Co-Infection and Trans-Encapsidation of Viral RNA In Vitro Using Epitope-Tagged Foot-and-Mouth Disease Viruses"

_viruses, 2021, doi:10.3390/v13122433_

Round 1

Reviewer 1 Report

Childs et al., clearly show trans-encapsidation of viral RNA into epitope tagged virus particles in vitro. The authors used the flexibility and plasticity of the GH loop in FMDV VP1 to introduce HA and FLAG tags into the FMDV capsid and showed co-infection via immunofluoresence microscopy and flow cytometry. Childs et al., went on to show that chimaeric particles containing both HA and FLAG-tagged VP1 proteins could be isolated from co-infected cells and went on to show trans-encapsidation of viral genome from the co-infecting virus. The authors have described a highly useful system for the study of co-infection in vitro without the need for sero-specific antibodies and has the potential to be a powerful tool for the analysis of recombination events between closely related virus strains. I would like to congratulate the authors on a well-designed study which with a few modifications should be ready for publication. All recommendations are minor in nature but could help improve the manuscript for the reader.

  1. Figure 1 - a schematic highlighting the position of the tag insertion relative to P1/VP1/RGD/2A would be helpful for the reader

  2. Figure 1a + 3a – If the authors could also add a loading control to the immunoblot panel that would be beneficial to confirm that similar MOIs are beign compared.

  3. Figure 1b -  It would be helpful if there was a larger field of view of the infected cell monolayer to show a number of infected cells then used the inlet panel to show a zoomed in image of single cells including the co-infected cells already shown in Figure 1b.

  4. Figure 2 + 3b – There appears to be large discrprencies in the percentage of infected cells when comparing the single infection of either HA or FLAG-tagged viruses in both figures. I was wondering if the authors could refer to this in there discussion of the results as to why this might be when the MOI used is so similar.  Additionally, the percentages given here appear to represent a n=1 one experiment, is it possible to highlight how often this experiment has happened to add an error rate (+/- % infected cells)

  5. Could the authors clarify why the MOI of 1.0 and 0.5 per virus in the co-infection was used as this leads to a low level of co-infected cells which potentially makes this harder to study, could this MOI be increased?

  6. Figure 4b could be described more cleary in the text to highlight what result you would expect to see, in particular it would be helpful if the authors could refer to the IgG band in the text as this would help the reader to interpret the figure more clearly. I would also like to commend the authors on Figure 4c as this is described very clearly

Reviewer 2 Report

Childs and colleagues using a previously described approach of tagging FMDV capsid protein VP1 with either HA or FLAG epitopes investigated the formation of chimeric capsids upon co-infection of cells with different viral strains. Using immunofluorescence and FACS analysis they demonstrated presence of both antigens in co-infected cells, and co-immunoprecipitation convincingly showed the formation of chimeric capsids.   The viruses and methods described in this paper may be useful for detailed investigation of recombination and trans-encapsidation of FMDVs, which has important implications for the development of vaccine-resistant recombinants and the spread of infection. The experiments are performed on high technical level and well presented.

I, however, cannot agree with the interpretation of the results presented on Fig. 4C (PCR) as evidence of trans-encapsidation. Strictly speaking, trans-encapsidation implies packaging of the viral genome in the capsid provided by the other virus (strain). These result does not show that the HA-encoding RNA is encapsidated entirely in FLAG-derived capsid and vice versa. They only provide additional evidence for the formation of mixed capsids containing both HA and FLAG-tagged units. To evaluate the level of true trans-encapsidation, a more detailed investigation of the progeny virions and the packaged RNAs is required.

Minor comment:

Lines 210, 238 MOI should be indicated in the text, not only in the Figure legend. 

Reviewer 3 Report

The authors are requested to explain why the goat epithelial cells were selected for the described studies? Do the authors think that the results would have been different in cells derived from another species and/or another cell type?

  1. The authors have described a method that allowed them to investigate whether heterologous FMDV capsids can be generated upon co-infection of two different FMDV strains. And the authors could also address the question whether trans-encapsidation of genomic RNA can occur.
  2. The manuscript is well-written, and the methods and results are clearly described. As far as this reviewer can tell, there are no typos or errors in the manuscript. The approach and scientific data is sound.
  1. There is one point of concern (minor point for improvement): The authors are requested to explain why the goat epithelial cells were selected for the described studies? Do the authors think that the results would have been different in cells derived from another species and/or another cell type? The authors should address this point in the manuscript, either by providing additional data or by addressing it in for instance the Discussion section.

Reviewer 4 Report

Foot-and-mouth disease virus (FMDV) belongs to the family Picornaviridae and a critical infectious agent to several important livestock species. Since antibodies induced by infection or vaccination are generally strain-specific and their cross-neutralizing activity is very poor, emerging strains can be a threat. Also, co-infection with antigenically diverse strains has a potential problem to generate novel high pathogenic viruses, suggesting the importance of study on co-infection dynamics and trans-encapsidation of different strains of FMDV.

This study described that co-infection with different serotypes of FMDV can cause production of chimeric capsids and trans-encapsidation of viral genomes, using different epitope-tagged recombinant viruses. Additionally, the authors suggested that detecting epitope-tagged VP1 proteins can be a useful strategy to study co-infection with the same or different strains of FMDV.

Overall, this paper provides limited novel information. Although the authors firstly demonstrated trans-encapsidation of FMDV genome from co-infected viruses, this phenomenon has been reported in other viruses too. So, the results are somehow predictable. However, the paper is clearly written and most of the results have been clearly presented. There are some aspects that need additional clarifications (see specific comments).

Specific Comments:

1) In figure 3B, HA-FMDV Asia 1/O1K shows higher infectivity compared to HA- and FLAG-FMDV O1K/O UKG35. If the authors generated it as a new recombinant strain, it would be important to provide experimental data that characterize this strain using a standard multi-step growth kinetics.

Round 2

Reviewer 4 Report

I think that the revised manuscript has been sufficiently improved for publication.